# Ex vivo lung CT findings may predict the outcome of the early phase after lung transplantation

Hisashi Oishi[1]*, Masafumi Noda[1], Tetsu Sado[1], Yasushi Matsuda[2], Hiromichi Niikawa[1], Tatsuaki Watanabe[1], Akira Sakurada[1], Yasushi Hoshikawa[2], Yoshinori Okada[1]

1 Department of Thoracic Surgery, Institute of Development, Aging and Cancer, Tohoku University, Sendai, Japan, 2 Department of Thoracic Surgery, Fujita Health University School of Medicine, Toyoake, Japan

* hisashi.oishi.c7@tohoku.ac.jp

## Abstract

### Purpose

We developed an ex vivo lung CT (EVL-CT) technique that allows us to obtain detailed CT images and morphologically assess the retrieved lung from a donor for transplantation. After we recovered the lung graft from a brain-dead donor, we transported it to our hospital and CT images were obtained ex vivo before lung transplant surgery. The objective of this study was to investigate the correlation between the EVL-CT findings and post-transplant outcome in patients who underwent bilateral lung transplantation (BLT) or single lung transplantation (SLT).

### Methods

We retrospectively reviewed the records of 70 patients with available EVL-CT data who underwent BLT (34 cases) or SLT (36 cases) in our hospital between October 2007 and September 2017. The recipients were divided into 2 groups (control group, infiltration group) according to the findings of EVL-CT of the lung graft in BLT and SLT, respectively. Recipients in the control group were transplanted lung grafts without any infiltrates (BLT control group, SLT control group). Recipients in the infiltration group received lung grafts with infiltrates (BLT infiltration group, SLT infiltration group).

### Results

The recipients in the BLT infiltration group showed significantly slower recovery from primary graft dysfunction and a longer mechanical ventilation period and ICU stay period than those in the BLT control group. The mechanical ventilation period was significantly longer in the recipients in the SLT infiltration group than those in the SLT control group.

### Conclusion

EVL-CT may predict the outcome of the early phase after lung transplantation.

**Data Availability Statement:** All relevant data are within the paper.

**Funding:** The authors received no specific funding for this work.

**Competing interests:** The authors have declared that no competing interests exist.

## Introduction

Lung transplantation is an effective therapeutic option for patients with end-stage lung diseases. However, the 5-year survival rate for adult lung transplant recipients is approximately 60%, according to the registry report of the International Society for Heart and Lung Transplantation (ISHLT). [1] The survival rate is still lower compared to the transplantation of other organs. We often must utilize lung grafts from marginal donors with localized pneumonia or lung contusion because we are faced with a severe donor shortage. [2] Chest CT scans of such donors are important for the assessment of the lung graft and useful for subsequent postoperative management of the recipient. However, chest CT scans taken immediately before the lung retrieval are not available in most cases because it is not always possible to take chest CT scans of a brain-dead donor at the hospital.

Verleden et al. reported CT scans of frozen whole lungs that were declined for transplantation due to allograft-related or non–allograft-related reasons. Interestingly, in their study, 4 of 8 cases showed CT alterations, whereas they were declined due to non-allograft-related reasons. [3] We developed an ex vivo lung CT (EVL-CT) technique that can be performed as part of the clinal practice because it is very simple and requires only a few minutes. This technique allows us to obtain detailed CT images and morphologically assess the already retrieved lung using an ex vivo method. However, the usefulness of EVL-CT in lung transplantation is still unknown. The objective of this study was to investigate correlations between the EVL-CT findings and post-transplant outcome in patients who underwent bilateral lung transplantation (BLT) or single lung transplantation (SLT).

## Patients and methods

### EVL-CT technique

When we recovered a lung graft from a brain-dead donor, we inflate it with a sustained airway pressure of 15–20 cmH$_2$O and a fraction of inspiratory oxygen of 50%. Then, we transport the lung graft to Tohoku University Hospital for the following lung transplant surgery. Immediately after arriving at the hospital, the lung graft is transferred to the medical imaging center and CT images of 1.25-mm-thick slice are obtained by BrightSpeed Elite (GE Healthcare Japan Ltd, Tokyo, Japan). Routine helical CT scans of whole lung are performed at a peak tube voltage of 120 kVp, with a variable mAs setting using an automatic exposure control system (Fig 1A and 1B). Only 3 minutes are required for EVL-CT.

After EVL-CT, lungs are implanted into the recipients by the usual surgical procedure. When we performed EVL-CT in the present study, the recipient was already under anesthesia and the lung transplant surgery was ongoing. In our current practice, we do not decide the availability of the lung graft based on the EVL-CT data.

### Patients and study groups

EVL-CT data from cases since October 2007 are available. Since then, EVL-CT was done in all lung transplant cases except living-donor lung transplant cases. We retrospectively reviewed the records of 92 patients who underwent lung transplantation between October 2007 and September 2017. We excluded living-donor lung transplantations (10 cases), lobar lung transplantations (6 cases) and retransplantations (6 cases) from this study. Ultimately, 70 patients with available EVL-CT data who underwent BLT or SLT were included in this study. Thirty-four BLT cases and 36 SLT cases were included (Fig 2).

The recipients were divided into 2 groups according to the findings of EVL-CT of the lung graft in SLT and BLT, respectively. Recipients in the control group were transplanted lung

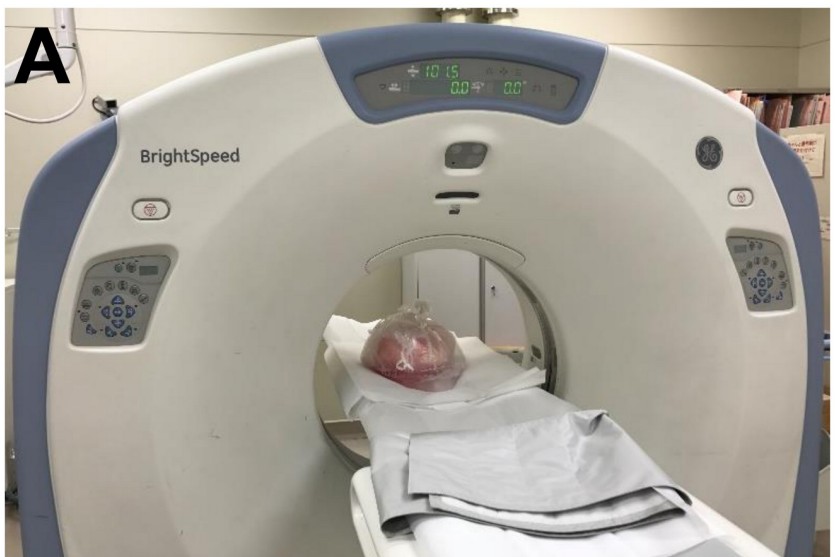

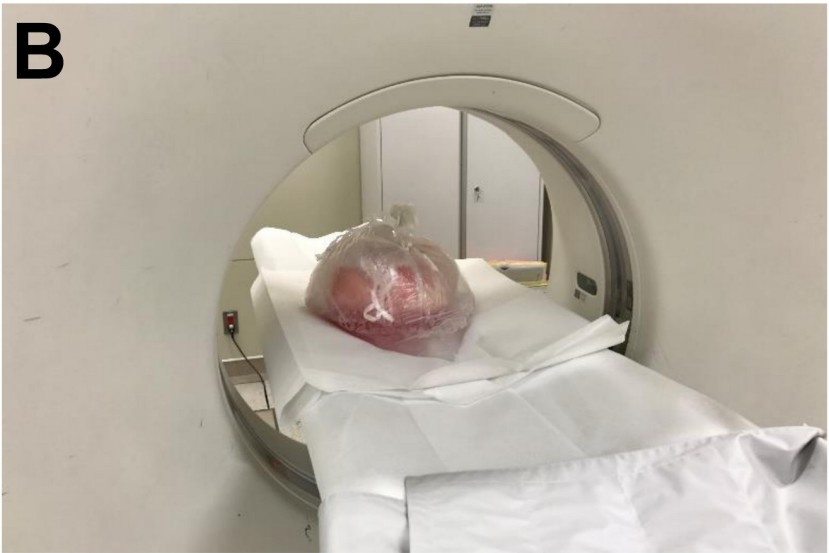

**Fig 1. Ex vivo donor lung CT technique.** After the retrieval of the lung graft from a brain-dead donor, we transported it to our hospital. Immediately after arriving at the hospital, the lung graft was transferred to the medical imaging center and CT images were obtained. (A) Overall view of the CT machine and the lung graft. (B) Close view of the lung graft in a plastic bag on the CT table.

grafts without any infiltrates (control group, Fig 3A, 3B and 3C). Recipients in the infiltration group received lung grafts with infiltrates (infiltration group, Fig 4A, 4B, 4C and 4D). The CT readers were trained using images of typical cases from a pilot study. EVL-CT images of each case were interpreted by two radiologists and a thoracis surgeon (H.O.) who were blinded to the donor data. With a consistent diagnosis by 2 of 3 CT readers, the recipient was divided into the control or infiltration group.

The other collected data included the demographics of donors and the pre-transplant demographics of the recipients, surgical characteristics, period of mechanical ventilation and

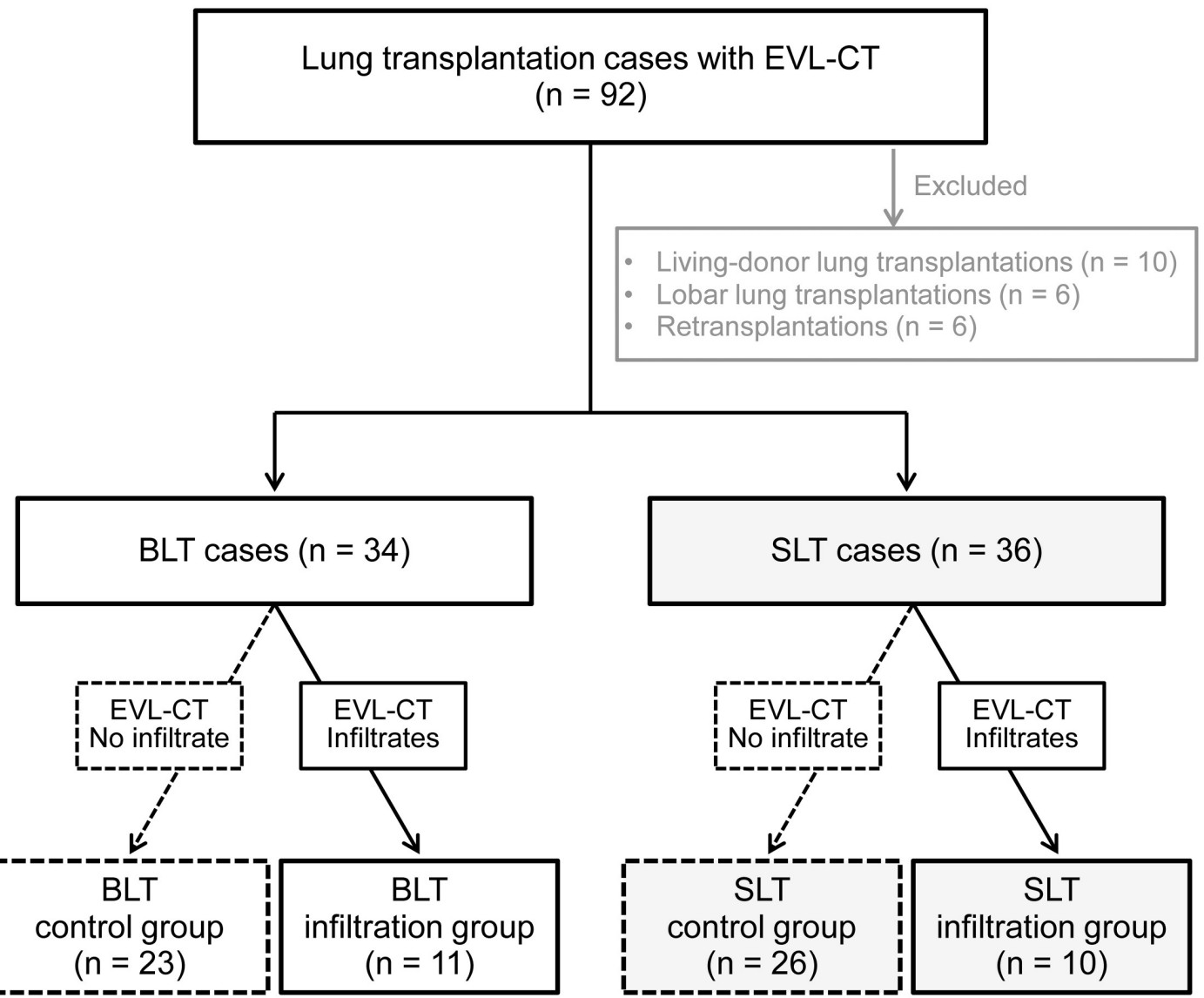

**Fig 2. Consort flow diagram indicating subject inclusion.** EVL-CT, ex vivo lung CT. BLT, bilateral lung transplantation. SLT, single lung transplantation.

ICU stay, complications, morbidity, mortality and the survival rate after lung transplantation. The data were compared between the control and the infiltration group in SLT and BLT, respectively. In the BLT cases, primary graft dysfunction (PGD) was graded according to the International Society for Heart and Lung Transplantation classification (ISHLT). [4] (PGD was not graded in the SLT cases because PGD grading with the classification is not always accurate in SLT cases due to the function of the native lung on the other side).

Japan Organ Transplant Network obtained consent for the recovery of lung grafts. If the individual's intentions are unclear, his/her organs can now be donated with family consent. The Institutional Review Board of Tohoku University Hospital approved the study (No. 2018-1-125) and the research was conducted in accordance with the 2000 Declaration of Helsinki

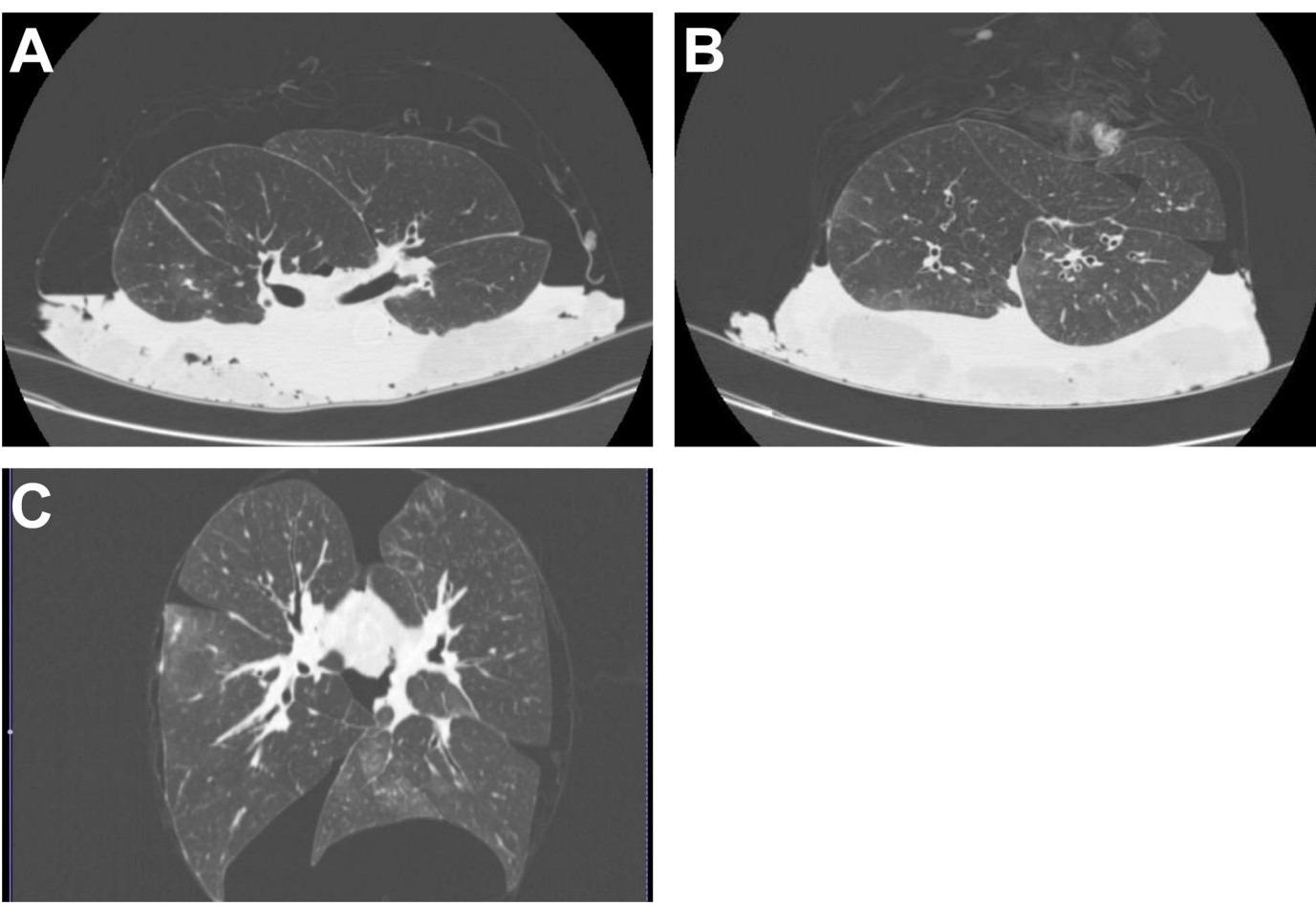

**Fig 3. Ex vivo donor lung CT images of a case from the bilateral lung transplantation (BLT) control group.** CT images show no infiltrate in the lung graft. (A, B) Transaxial CT images of the lung graft. (C) A coronal image of the lung graft.

and the Declaration of Istanbul 2008. All patients gave written informed consent for EVL-CT and data collection.

## Statistical analysis

Data are expressed as mean ± standard deviation for normally distributed data and Student's *t*-test was used. Data are expressed as median with range for non-normally distributed data and the Mann-Whitney *U* test was used. Fisher's exact test or the chi-square test was performed for categorical values. The PGD grade at each time point was analyzed by repeated measures two-way ANOVA. Prism 5 (GraphPad Software Inc., La Jolla, CA) was used to perform these statistical analyses. Values of $p < 0.05$ were regarded as significant. We did not perform sample size calculations because the present study is an exploratory research.

## Results

In 23 of 34 BLT cases (67.6%), EVL-CT showed no infiltrate in the lung graft (BLT control group). On the other hand, EVL-CT displayed infiltrates in the lung graft in 11 cases (32.4%,

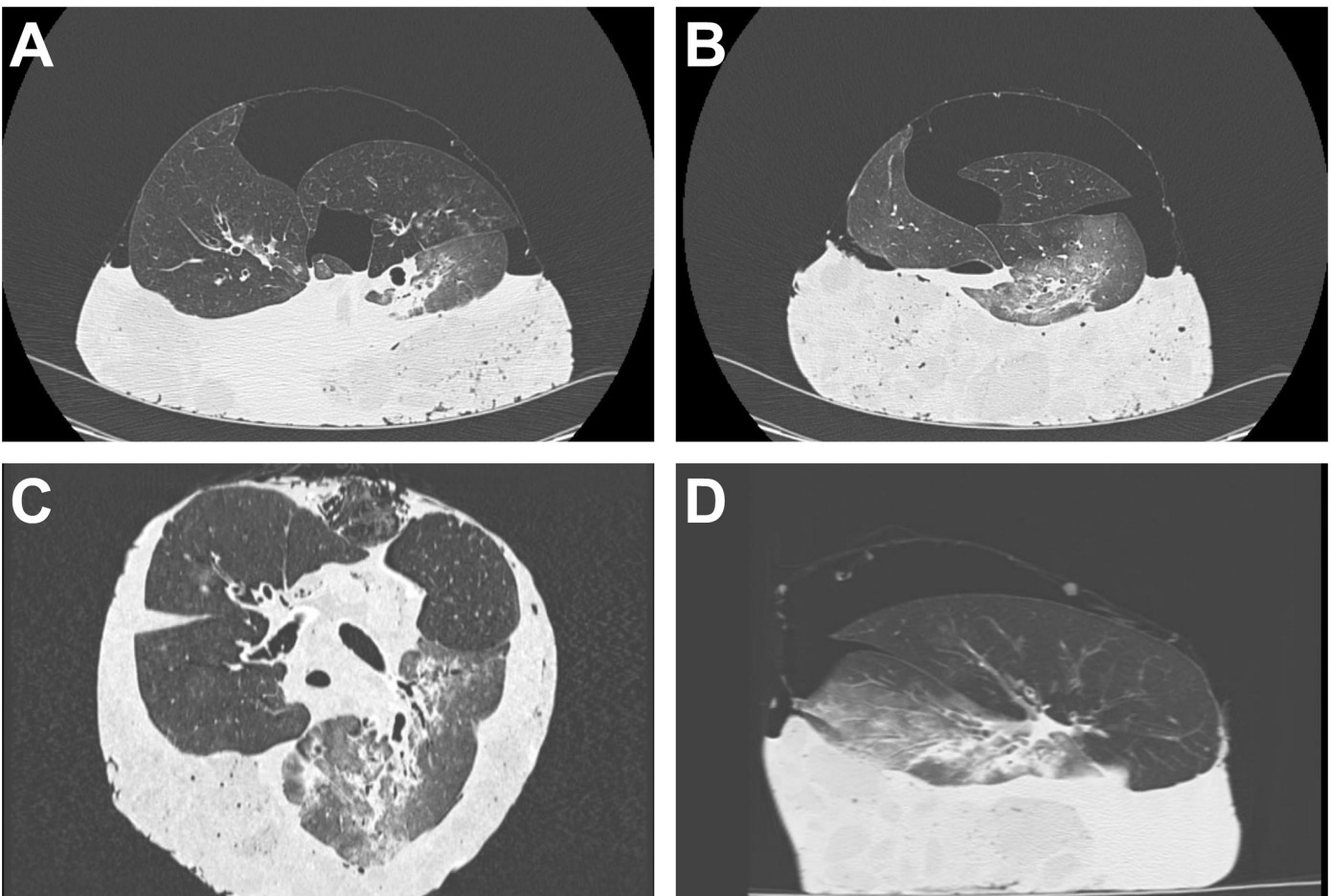

**Fig 4. Ex vivo donor lung CT images of a case from the bilateral lung transplantation (BLT) infiltration group.** CT images show infiltrates in the left lower lobe of the lung graft. (A, B) Transaxial CT images of the lung graft. (C) A coronal image of the lung graft. (D) A sagittal image of the lung graft.

BLT infiltration group). There were 26 cases in the SLT control group (72.2%) and 10 cases in the SLT infiltration group (27.8%) (Fig 2). Structural lung diseases, such as emphysema and interstitial lung disease, were not detected in any cases in the present study.

Table 1 shows the demographics of the donors in each group in BLT. In most of the cases (73.9% in the BLT control group, 72.7% in the BLT infiltration group), there was no available chest CT image within 3 days before retrieval. Preretrieval CT showed infiltrates in the lung graft in 4 cases in the BLT control group and these infiltrates all seem to be small atelectasis. There was no significant difference in any parameters between the two groups except in the bronchial aspirates in the culture. The percentage of positive bronchial aspirates in the culture was significantly higher in the BLT infiltration group. Methicillin-susceptible staphylococcus aureus was the most frequently detected microorganism in the BLT infiltration group.

Table 2 shows the demographics of the donors in each group in SLT. Similarly to the cases in BLT, in most of the cases (65.4% in the SLT control group, 80.0% in the SLT infiltration group) there was no available chest CT image within 3 days before retrieval. The percentage of positive bronchial aspirates in the culture was significantly higher in the SLT infiltration group.

**Table 1. Demographics of the donors in bilateral lung transplantation.**

| | BLT[a] control group (N = 23) | BLT infiltration group (N = 11) | *P* value |
|---|---|---|---|
| Donor age (years) | 43.7 ± 13.5 (6–63) | 38.4 ± 17.1 (11–62) | 0.35 |
| Donor gender (M/F) | 10 / 13 | 6 / 5 | 0.72 |
| Donor height (cm) | 161.2 ± 11.2 (125–175) | 165.7 ± 10.3 (150–184) | 0.29 |
| Donor weight (kg) | 56.8 ± 12.3 (37.0–67.7) | 62.6 ± 17.4 (43–73.3) | 0.29 |
| Donor BMI (kg/m$^2$)[b] | 21.6 ± 3.7 (12.8–31.6) | 22.5 ± 4.5 (18.1–34.6) | 0.58 |
| Preretrieval CXR[c] findings | | | 0.08 |
| Infiltration | 3 (13.0%) | 5 (45.5%) | |
| No infiltration | 20 (87.0%) | 6 (54.5%) | |
| Preretrieval CT images[d] | | | 1.00 |
| Unavailable | 17 (73.9%) | 8 (72.7%) | |
| Available | 6 (26.1%) | 3 (27.3%) | |
| Infiltration | 4 (66.7%) | 3 (100%) | |
| No infiltration | 2 (33.3%) | 0 (0%) | |
| Smoking history (pack-years) | | | 0.39 |
| None | 11 (47.8%) | 7 (63.6%) | |
| 0–20 | 7 (30.4%) | 1 (9.1%) | |
| 20 ≤ | 5 (21.7%) | 3 (27.3%) | |
| PaO$_2$ / FiO$_2$ (mmHg) | 510 ± 80 (293–612) | 478 ± 67 (335–545) | 0.27 |
| Cause of brain death | | | 0.83 |
| Cerebrovascular accident | 11 (47.8%) | 5 (45.5%) | |
| Head trauma | 7 (30.4%) | 3 (27.3%) | |
| Brain ischemia | 4 (17.4%) | 3 (27.3%) | |
| Others | 1 (4.3%) | 0 (0%) | |
| Bronchial aspirates culture | | | **0.004** |
| Negative | 17 (73.9%) | 2 (18.2%) | |
| Positive | 6 (26.1%) | 9 (81.8%) | |
| MSSA[e] | 1 (4.3%) | 4 (36.4%) | |
| Candida species | 2 (8.7%) | 3 (27.3%) | |
| Coagulase-negative staphylococci | 0 (0%) | 2 (18.2%) | |
| Pseudomonas aeruginosa | 0 (0%) | 2 (18.2%) | |
| Others | 3 (13.0%) | 2 (18.2%) | |

Data are expressed as group mean ± standard deviation or number (%).

[a]BLT, bilateral lung transplantation.

[b]BMI, body mass index.

[c]Chest X-ray.

[d]CT images taken within 3 days.

[e]MSSA, methicillin-susceptible staphylococcus aureus.

Tables 3 and 4 show the pre-transplant demographics of the recipients in each group in BLT or SLT, respectively. There was no significant difference in any parameters between the two groups in both BLT and SLT. Tables 5 and 6 show the surgical characteristics in each group in BLT or SLT, respectively. Operative time, cold ischemic time and extracorporeal circulation use were similar between the control and infiltration groups in both BLT and SLT. The amount of intra-operative blood loss tended to be larger in the infiltration group in BLT.

**Table 2. Demographics of the donors in single lung transplantation.**

| | SLT[a] control group (N = 26) | SLT infiltration group (N = 10) | *P* value |
|---|---|---|---|
| Donor age (years) | 45.7 ± 12.6 (24–68) | 40.0 ± 12.6 (18–56) | 0.24 |
| Donor gender (M/F) | 7 / 19 | 6 / 4 | 0.12 |
| Donor height (cm) | 158.8 ± 7.4 (147–177) | 162.4 ± 6.7 (151–170) | 0.25 |
| Donor weight (kg) | 51.3 ± 12.8 (31.6–73.0) | 49.6 ± 13.3 (32.3–69.7) | 0.73 |
| Donor BMI[b] (kg/m$^2$) | 20.4 ± 5.3 (13.7–31.1) | 18.6 ± 4.0 (13.3–24.8) | 0.35 |
| Preretrieval CXR[c] findings | | | **0.002** |
| Infiltration | 2 (7.7%) | 6 (60.0%) | |
| No infiltration | 24 (92.3%) | 4 (40.0%) | |
| Preretrieval CT images[d] | | | 0.69 |
| Unavailable | 17 (65.4%) | 8 (80.0%) | |
| Available | 9 (34.6%) | 2 (20.0%) | |
| Infiltration | 0 (0%) | 2 (100.0%) | |
| No infiltration | 9 (100.0%) | 0 (0%) | |
| Smoking history (pack-years) | | | 0.89 |
| None | 11 (42.3%) | 5 (50.0%) | |
| 0–20 | 10 (38.5%) | 3 (30.0%) | |
| 20 ≤ | 5 (19.2%) | 2 (20.0%) | |
| PaO$_2$ / FiO$_2$ (mmHg) | 464 ± 97 (237–606) | 423 ± 116 (187–576) | 0.27 |
| Cause of brain death | | | |
| Cerebrovascular accident | 14 (53.8%) | 7 (70.0%) | |
| Brain ischemia | 8 (30.8%) | 2 (20.0%) | |
| Head trauma | 1 (3.8%) | 1 (10.0%) | |
| Others | 3 (11.6%) | 0 (0%) | |
| Bronchial aspirates culture | | | **0.001** |
| Negative | 21 (80.8%) | 2 (20.0%) | |
| Positive | 5 (19.2%) | 8 (80.0%) | |
| MSSA[e] | 1 (3.8%) | 4 (40.0%) | |
| Candida species | 3 (11.5%) | 3 (30.0%) | |
| Klebsiella pneumonia | 0 (0%) | 3 (30.0%) | |
| MRSA[f] | 1 (3.8%) | 2 (20.0%) | |
| Corynebacterium species | 2 (7.7%) | 0 (0%) | |
| Others | 3 (11.5%) | 5 (50.0%) | |

Data are expressed as group mean ± standard deviation or number (%).

[a]SLT, single lung transplantation.

[b]BMI, body mass index.

[c]Chest X-ray.

[d]CT images taken within 3 days.

[e]MSSA, methicillin-susceptible staphylococcus aureus.

[f]MRSA, methicillin-resistant staphylococcus aureus.

Fig 5 shows the EVL-CT and post-transplant chest CT images of a typical lung transplant case in the BLT infiltration group. The recipient was diagnosed with emphysema and was on the waitlist for lung transplantation for 3 years. The lung graft was donated from a brain-dead donor with intracerebral hemorrhage. No infiltrate was mentioned in the chest X-ray of the donor taken immediately before retrieval. However, the EVL-CT of the graft demonstrated

**Table 3. Pre-transplant demographics of recipients in bilateral lung transplantation.**

| | BLT[a] control group (N = 23) | BLT infiltration group (N = 11) | *P* value |
|---|---|---|---|
| Time on waitlist (days) | 1180 ± 869 (11–4081) | 1137 ± 538 (470–2016) | 0.89 |
| Age (years) | 37.4 ± 13.5 (14–55) | 39.9 ± 9.5 (22–51) | 0.62 |
| Gender (M/F) | 9 / 14 | 5 / 6 | 1.00 |
| Height (cm) | 158.5 ± 10.0 (135–170) | 162.3 ± 5.7 (155–176) | 0.27 |
| Weight (kg) | 45.4 ± 9.3 (29–58) | 51.7 ± 12.7 (37–77) | 0.16 |
| BMI (kg/m$^2$)[b] | 18.0 ± 3.3 (12.3–25.4) | 19.5 ± 4.0 (13.6–27.7) | 0.28 |
| Indication | | | 0.94 |
| IPAH[c] | 8 (34.8%) | 3 (27.3%) | |
| Bronchiectasis and DPB[d] | 5 (21.7%) | 3 (27.3%) | |
| PH-not IPAH[e] | 3 (13.0%) | 2 (18.2%) | |
| Others | 7 (30.4%) | 3 (27.3%) | |

Data are expressed as group mean ± standard deviation or number (%).

[a]BLT, bilateral lung transplantation.

[b]BMI, body mass index.

[c]IPAH, idiopathic pulmonary arterial hypertension.

[d]DPB, diffuse panbronchiolitis.

[e]PH-not IPAH, pulmonary hypertension that is not IPAH.

infiltrates in the left lower lobe (Fig 5A and 5B). The bronchial aspirates in the culture of the donor were positive for pseudomonas aeruginosa. We presumed that the donor had pneumonia caused by pseudomonas aeruginosa. One week after the transplant, the chest CT scans of the recipients revealed that infiltrates remained in the left lower lobe (Fig 5C). Because we

**Table 4. Pre-transplant demographics of recipients in single lung transplantation.**

| | SLT[a] control group (N = 26) | SLT infiltration group (N = 10) | *P* value |
|---|---|---|---|
| Time on waitlist (days) | 946 ± 658 (193–3335) | 892 ± 531 (252–1878) | 0.82 |
| Age (years) | 45.4 ± 10.1 (29–61) | 47.1 ± 10.4 (23–61) | 0.67 |
| Gender (M/F) | 5 / 21 | 5 / 5 | 0.10 |
| Height (cm) | 158.8 ± 7.4 (147–177) | 162.4 ± 6.7 (151–170) | 0.19 |
| Weight (kg) | 51.4 ± 12.7 (31.6–73) | 49.6 ± 13.3 (32.3–69.7) | 0.71 |
| BMI (kg/m$^2$)[b] | 20.5 ± 5.3 (13.7–31.1) | 18.6 ± 4.0 (13.3–24.8) | 0.33 |
| Indication | | | 0.54 |
| LAM[c] | 15 (57.7%) | 4 (40.0%) | |
| IPF[d] | 4 (15.4%) | 3 (30.0%) | |
| COPD[e] | 2 (7.7%) | 1 (10.0%) | |
| CTD-IP[f] | 2 (7.7%) | 1 (10.0%) | |
| Others | 3 (11.5%) | 1 (10.0%) | |

Data are expressed as group mean ± standard deviation or number (%).

[a]SLT, single lung transplantation.

[b]BMI, body mass index.

[c]LAM, lymphangioleiomyomatosis.

[d]IPF, idiopathic pulmonary fibrosis.

[e]COPD, chronic obstructive pulmonary disease.

[f]CTD-IP, collagen tissue disease-associated interstitial pneumonia.

**Table 5. Surgical characteristics in bilateral lung transplantation.**

| | BLT[a] control group (N = 23) | BLT infiltration group (N = 11) | *P* value |
|---|---|---|---|
| Operative time (min) | 904 ± 197 (576–1439) | 999 ± 238 (798–1531) | 0.24 |
| Cold ischemic time | | | |
| 1st lung (min) | 543 ± 134 (244–792) | 557 ± 87 (473–785) | 0.76 |
| 2nd lung (min) | 701 ± 91 (550–870) | 743 ± 45 (656–829) | 0.17 |
| Extracorporeal circulation use | | | |
| CPB[b] or CPB + ECMO | 14 (60.9%) | 6 (54.5%) | |
| ECMO[c] | 6 (26.1%) | 5 (45.5%) | |
| No | 3 (13.0%) | 0 (0%) | |
| Intra-operative blood loss (ml) | 6841 ± 5537 | 13666 ± 14681 | 0.07 |
| | (289–15672) | (837–47429) | |

Data are expressed as group mean ± standard deviation or number (%).

[a]BLT, bilateral lung transplantation.

[b]CPB, cardio-pulmonary bypass.

[c]ECMO, extracorporeal membrane oxygenation.

already knew that infiltrates resulted from pneumonia of the donor, we continued the administration of effective antibiotics and the infiltrates gradually disappeared over time (Fig 5D, 5E and 5F).

In BLT, 11 of 23 (47.8%) patients were on extracorporeal membrane oxygenation (ECMO) on ICU arrival in the control group, 4 of 11 (36.4%) patients were on ECMO in the infiltration group with no significant difference between the 2 groups ($p = 0.72$). There was also no significant difference in ICU mortality between them (control group, 4 of 23, 17.4%; infiltration group, 1 of 11, 9.1%; $p = 1.00$). The mechanical ventilation period in the infiltration group was significantly longer than that in the control group (19.9 ± 12.9 vs. 41.9 ± 27.3 days, $p = 0.009$; Fig 6A). The ICU stay period in the infiltration group was also significantly longer than that in the control group (28.2 ± 15.0 vs. 54.3 ± 27.5 days, $p = 0.004$; Fig 6B). ICU mortality cases (control group, 4 cases; infiltration group, 1 case) were excluded from each group.

**Table 6. Surgical characteristics in single lung transplantation.**

| | SLT[a] control group (N = 26) | SLT infiltration group (N = 10) | *P* value |
|---|---|---|---|
| Operative time (min) | 405 ± 89 (243–598) | 411 ± 61 (298–503) | 0.84 |
| Cold ischemic time (min) | 457 ± 48 (389–572) | 477 ± 65 (361–583) | 0.34 |
| Extracorporeal circulation use | | | |
| CPB[b] | 1 (3.8%) | 0 (0%) | |
| ECMO[c] | 14 (53.8%) | 6 (60.0%) | |
| No | 11 (42.3%) | 4 (40.0%) | |
| Intra-operative blood loss (ml) | 1165 ± 1995 | 2102 ± 2823 | 0.29 |
| | (116–10354) | (79–8750) | |

Data are expressed as group mean ± standard deviation or number (%).

[a]SLT, single lung transplantation.

[b]CPB, cardio-pulmonary bypass.

[c]ECMO, extracorporeal membrane oxygenation.

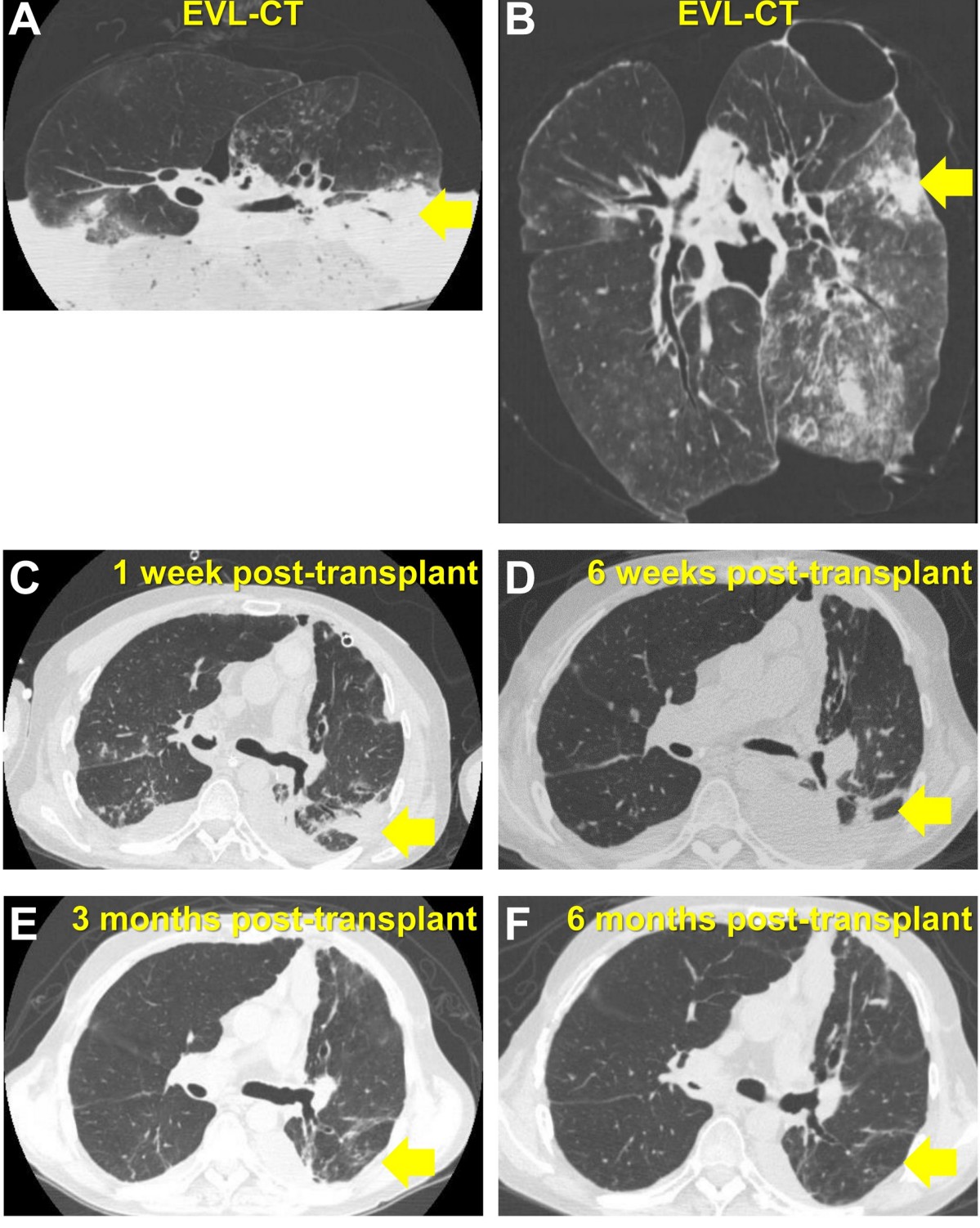

**Fig 5. Ex vivo donor lung CT and post-transplant chest CT images of a typical lung transplant case in the bilateral lung transplantation (BLT) infiltration group.** (A, B) Transaxial and coronal CT images of ex vivo lung CT (EVL-CT) show infiltrates in the left lower lobe. The arrow indicates infiltrates in the superior segment. (C) The infiltrates remained at one week post-transplant. (D, E, F) The infiltrates gradually disappeared over time.

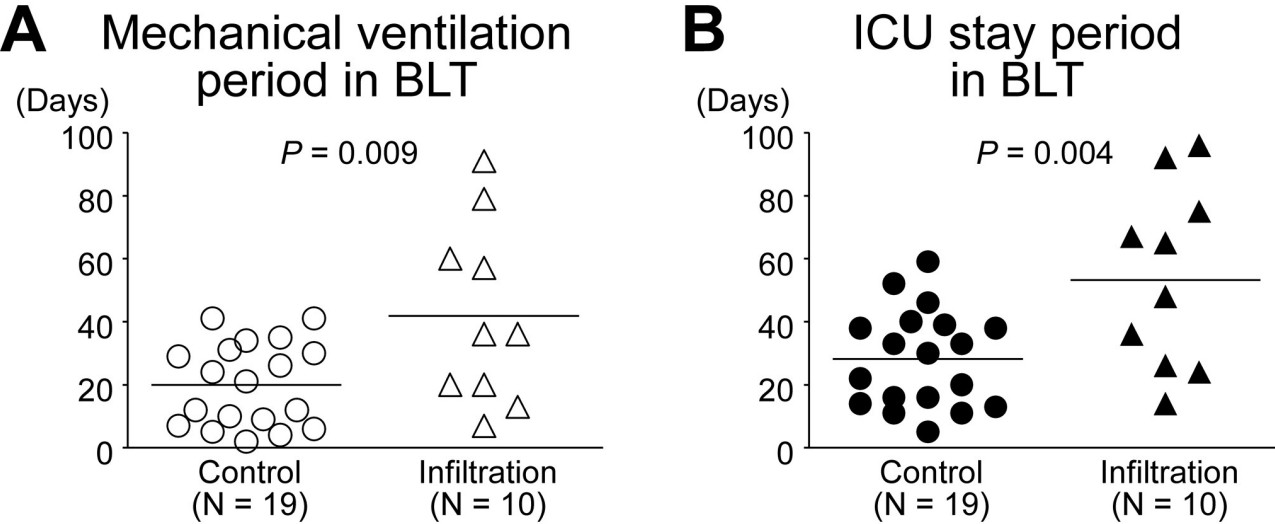

**Fig 6. Post-transplant mechanical ventilation period and ICU stay period in bilateral lung transplantation (BLT).** (A) Mechanical ventilation period (days) in each group. The mechanical ventilation period in the BLT infiltration group was significantly longer than that in the BLT control group. (B) ICU stay period (days) in each group. The ICU stay period in the infiltration group was also significantly longer than that in the BLT control group. ICU mortality cases (control group, 4 cases; infiltration group, 1 case) were excluded from each group.

In SLT, 6 of 26 (23.1%) patients were on ECMO on ICU arrival in the control group, 2 of 10 (20.0%) patients were on ECMO in the infiltration group, with no significant difference between the 2 groups ($p$ = 1.00). There was also no significant difference in ICU mortality between them (control group, 1 of 26, 3.8%; infiltration group, 1 of 10, 10.0%; $p$ = 0.48). The mechanical ventilation period in the infiltration group was significantly longer than that in the control group (6.2 ± 5.7 vs. 11.9 ± 7.9 days, $p$ = 0.03; Fig 7A). The ICU stay period in the

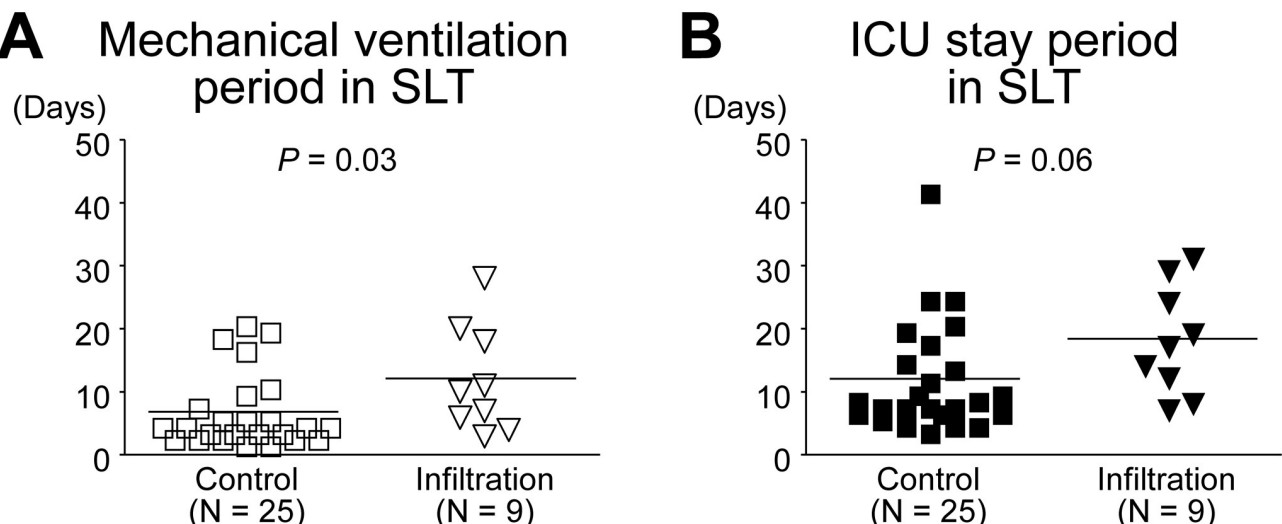

**Fig 7. Post-transplant mechanical ventilation period and ICU stay period in single lung transplantation (SLT).** (A) Mechanical ventilation period (days) in each group. The mechanical ventilation period in the SLT infiltration group was significantly longer than that in the control group. (B) ICU stay period (days) in each group. The ICU stay period in the infiltration group tended to be longer than that in the control group; however, the difference was not significant. ICU mortality cases (control group, 1 case; infiltration group, 1 case) were excluded from each group.

## Proportion of PGD 1-3 patients

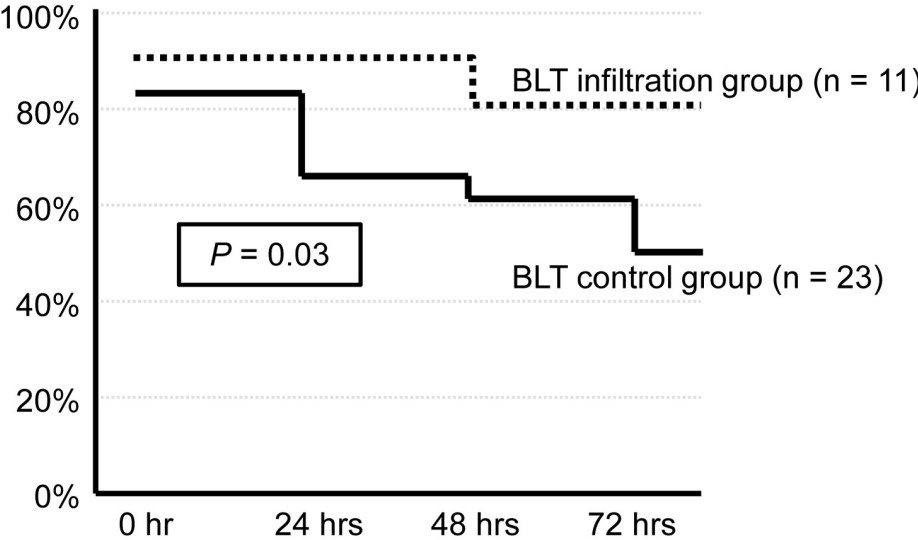

**Fig 8. Primary graft dysfunction grade in bilateral lung transplantation (BLT).** Primary graft dysfunction (PGD) was graded according to the international society for heart and lung transplantation classification (ISHLT). [4] Grade 0 indicates the ratio of the partial pressure of arterial oxygen to the fraction of inspired oxygen (PaO$_2$:FiO$_2$) ≥300 mm Hg with clear chest radiographs; grade 1 PaO2:FiO2 ≥300 mm Hg with infiltration on chest radiographs; grade 2 PaO2:FiO2 ≥200 but <300 mm Hg; and grade 3 PaO2:FiO2 <200 mm Hg. This graph shows the proportion of patients with PGD grades 1 to 3 over time (first 3 days after transplantation) in BLT. The proportion of patients with PGD grades 1 to 3 in the infiltration group was significantly higher than that in the BLT control group ($p$ = 0.03).

infiltration group tended to be longer than that in the control group; however, the difference was not significant (12.3 ± 8.6 vs. 18.9 ± 8.2 days, $p$ = 0.06; Fig 7B).

Fig 8 shows the proportion of patients with PGD grades 1 to 3 in the first 3 days after transplantation. The proportion of patients with PGD grades 1 to 3 in the BLT infiltration group was significantly higher than that in the BLT control group ($p$ = 0.03). In other words, the improvement of the PGD grades in the BLT control group was significantly faster than that in the BLT infiltration group.

## Discussion

One of the ways to solve the problem of the waitlist mortality in lung transplantation is to maximize the lung utilization rate from marginal donors, such as donors with localized pneumonia or lung contusion. Moreno et al. reported that the use of extended criteria donors, including donors with pulmonary infiltrates on chest radiograph, did not increase the incidence of PGD or 30-day mortality. [5] Sundaresan et al. described that a mild infiltrate in one lung may be acceptable for bilateral lung transplants. [6] However, we must carefully evaluate the lung graft for a good outcome of the recipient.

One of the best methods to evaluate the infiltration of the donor lung is to take CT scans of the chest of the donor. In some donor cases, no chest CT scans are available and only chest X-rays are taken. Gauthier et al suggested that chest CT imaging might be an important adjunct to conventional lung donor assessment criteria. [7] Hoetzenecker also recommended routine chest CT scans of every donor for better judgement of the donor organ quality. [8] In fact, we

can obtain CT scans of a potential lung donor and utilize the information accordingly. However, even if chest CT scans are available, these CT scans are not taken immediately before the retrieval in most cases. Such CT scans are important to detect structural lung diseases and decide the nonutilization of the lung graft. [7] On the other hand, we seldom obtain accurate information that reflects the acute abnormalities at the time of retrieval. In addition, whether the CT scans are taken immediately right before the retrieval or not, how specific CT findings in donors can affect recipient outcomes remain to be determined. [9]

In the present study, we demonstrated that EVL-CT provided us with detailed morphological information of the lung graft. Surprisingly, only in 5 of 11 cases (45.5%) in the BLT infiltration group and in 6 of 10 cases (60.0%) in the SLT infiltration group were infiltrates reported by chest X-rays of the donor before retrieval. The sensitivity of preretrieval chest X-ray was proven to be significantly low. In terms of specificity of preretrieval chest X-ray in the present study, in 20 of 23 cases (87.0%) of the BLT control group and in 24 of 26 cases (92.3%) of the SLT control group, no infiltrate was detected in the preretrieval chest X-ray. Vanstapel et al. demonstrated a histopathologic evaluation of the donor lungs declined for transplantation. They reported that 3 of 23 (13.0%) donor lungs that were not transplanted due to extrapulmonary causes displayed severe histologic abnormalities (pneumonia, emphysema) [10]. Similar to their study, 3 of 23 cases (13.0%) in the BLT control group and in 2 of 26 cases (7.7%) in the SLT control group showed infiltrations in EVL-CT in our study. We think that the preretrieval assessment of lung grafts by chest X-ray is unsatisfactory and EVL-CT may be one of the options to determine the utilization of the graft.

There are several reasons why EVL-CT revealed infiltrates in the lung graft. The most frequent reason in this study was pneumonia. We present a case in which EVL-CT revealed pneumonia in the donor (Fig 5). In this case, the information acquired in the EVL-CT was very useful for the subsequent postoperative management. Prior to our study, Verleden et al. reported the CT scans of frozen whole lungs revealed parenchymal infiltrates consistent with infection even in cases declined for transplantation due to non–allograft-related reasons. [3] Similar to their study, the EVL-CT in the present study allowed us to obtain additional information that had not be noticed upon the retrieval. In the present study, because EVL-CT was performed as a part of the clinical practice and all lungs were implanted to recipients, we were able to utilize the information for postoperative recipient care and investigate the outcome according to the results of EVL-CT.

One of the advantages of EVL-CT is that we can assess a lung graft that was adequately inflated on the retrieval and was without atelectasis. Indeed, whereas preretrieval CT images showed infiltrates in 4 of 6 donors in the BLT control group, the EVL-CT images revealed no infiltrates. Martens et al. states that current evaluation of donor lung quality at the time of the offer is often challenging and therefore the retrieval team should reevaluate the lungs when fully ventilated after the recruitment of atelectatic zones. [11] We believe that we can distinguish other types of infiltrates from atelectasis by taking EVL-CT of the donor lung because lungs are inflated well on the retrieval. In the present study, we took EVL-CT of the lung graft after arriving at our hospital and the transplant surgery was ongoing; therefore, the lung was implanted regardless the result of the EVL-CT. In the future, we could perform EVL-CT at the donor hospital and decide the acceptance of the lung graft with the data of donor arterial blood and the EVL-CT images.

Considering the severe condition and the long waiting time for the recipient, we sometimes must utilize lung grafts from a donor whose chest CT scans show infiltrates. Most of the donors in the infiltration group probably had mild pneumonia or acute bronchitis. In fact, bronchial aspirates in the culture were more often positive in the infiltration group than in the control group in both BLT and SLT. However, bronchial aspirates in the culture should be

carefully interpreted because they do not always reflect organisms of the lower respiratory tract. [12] Other reasons for the finding of infiltration in EVL-CT included contusions due to chest trauma, and such infiltrations can easily be distinguished by the medical history of the donor.

Egan et al. reported a system of ex vivo evaluation of human lungs for transplant suitability in 2006. [13] In their study of 6 cases, ex vivo CT scans were obtained from human lungs deemed unsuitable for transplant. They stated that CT scan is a means to diagnose infiltrates and other abnormalities that might exclude lungs from being considered appropriate for transplantation. [13] In the present study, we present a series of 70 clinical lung transplant cases and demonstrated the simplicity and feasibility of EVL-CT. We also showed that, even though the ICU mortality was comparable, the improvement of PGD was slower and the mechanical ventilation period and ICU stay were longer in the BLT recipients who received lung graft with findings of infiltrates in EVL-CT. In addition, we demonstrated that the mechanical ventilation period was longer in the SLT recipients who received lung grafts with findings of infiltrates in EVL-CT. We believe that EVL-CT may predict the outcome of the early phase after lung transplantation.

There are several limitations in the present study. First, it was a retrospective single-center analysis with a small number of patients. Second, the EVLP technique that is currently in clinical use in some countries can assess the lung graft function ex vivo. On the other hand, whereas the EVL-CT used in the present study can provide us with morphological information of the lung graft, it cannot evaluate the function of the lung graft. Third, this was a non-interventional study and we did not change the following clinical practice based on the result of EVL-CT. As mentioned in Patients and Methods, at present we do not decide the availability of the lung graft based on the EVL-CT data. When we performed EVL-CT in the present study, the transplant surgery was already underway. We were not able to decline to use the lung graft for the lung transplant. Fourth, we were not able to completely eliminate selection bias. Due to the small number of patients, it was impossible to adjust for differences between the groups, for example propensity score-matching analysis.

In the future, we might be able to perform lobectomy on the back table according to the EVL-CT data in order to remove localized pneumonia or contusions, and then complete the lobar lung transplantation. As mentioned above, another future option may be performing EVL-CT at the donor hospital and decide the acceptance of the lung graft and then we start the lung transplant surgery of the recipient.

In conclusion, the BLT recipients who received lung graft with the findings of infiltrates in EVL-CT showed slower recovery from PGD and longer periods of mechanical ventilation and ICU stay than the BLT recipients without infiltrate. The mechanical ventilation period was significantly longer in the SLT recipients who received lung graft with findings of infiltrates in EVL-CT than in the SLT recipients without infiltrate. EVL-CT may predict the outcome of the early phase after lung transplantation.

## Acknowledgments

The authors thank the radiological technologists of Tohoku University Hospital for their assistance with the collection of data. The authors would like to express their gratitude to Brent Bell for assistance in editing this manuscript.

## Author Contributions

**Conceptualization:** Hisashi Oishi, Yoshinori Okada.

**Data curation:** Hisashi Oishi, Masafumi Noda, Tetsu Sado, Yasushi Matsuda, Hiromichi Niikawa, Tatsuaki Watanabe, Akira Sakurada, Yasushi Hoshikawa.

**Formal analysis:** Hisashi Oishi, Masafumi Noda, Tetsu Sado, Yasushi Matsuda, Tatsuaki Watanabe, Akira Sakurada.

**Supervision:** Yoshinori Okada.

**Writing – original draft:** Hisashi Oishi, Akira Sakurada.

**Writing – review & editing:** Masafumi Noda, Tetsu Sado, Yasushi Matsuda, Hiromichi Niikawa, Tatsuaki Watanabe, Yasushi Hoshikawa, Yoshinori Okada.

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
