## [Decision Letter · Decision Letter 0]

16 Mar 2020

PONE-D-20-03435

Extracted donor lung CT findings may predict post-transplant outcome in lung transplantation

PLOS ONE

Dear Dr. Oishi,

Thank you for submitting your manuscript to PLOS ONE. After careful consideration, we feel that it has merit but does not fully meet PLOS ONE’s publication criteria as it currently stands. Therefore, we invite you to submit a revised version of the manuscript that addresses the points raised during the review process.

ACADEMIC EDITOR: 

Your MS has been highlighted by expert reviewers as novel and interesting, and therefore I invite you to thoroughly revise your paper according to the very helpful comments of my colleagues. Looking forward to seeing your revised MS.

We would appreciate receiving your revised manuscript by Apr 30 2020 11:59PM. To enhance the reproducibility of your results, we recommend that if applicable you deposit your laboratory protocols in protocols.io, where a protocol can be assigned its own identifier (DOI) such that it can be cited independently in the future. For instructions see: http://journals.plos.org/plosone/s/submission-guidelines#loc-laboratory-protocols

We look forward to receiving your revised manuscript.

Kind regards,

Frank JMF Dor, M.D., Ph.D., FEBS, FRCS

Academic Editor

PLOS ONE

Journal Requirements:

2. Please provide a sample size and power calculation in the Methods, or discuss the reasons for not performing one before study initiation.

a)    Please provide an amended Funding Statement that declares *all* the funding or sources of support received during this specific study (whether external or internal to your organization) as detailed online in our guide for authors at http://journals.plos.org/plosone/s/submit-now.  

b)    Please state what role the funders took in the study.  If any authors received a salary from any of your funders, please state which authors and which funder. If the funders had no role, please state: "The funders had no role in study design, data collection and analysis, decision to publish, or preparation of the manuscript."

Reviewers' comments:

Reviewer's Responses to Questions

**Comments to the Author**

1. Is the manuscript technically sound, and do the data support the conclusions?

Reviewer #1: Yes

Reviewer #2: Partly

Reviewer #3: Yes

2. Has the statistical analysis been performed appropriately and rigorously? 

Reviewer #1: Yes

Reviewer #2: Yes

Reviewer #3: No

3. Have the authors made all data underlying the findings in their manuscript fully available?

Reviewer #1: Yes

Reviewer #2: Yes

Reviewer #3: Yes

4. Is the manuscript presented in an intelligible fashion and written in standard English?

Reviewer #1: Yes

Reviewer #2: Yes

Reviewer #3: No

5. Review Comments to the Author

Reviewer #1: This is an interesting study, examining pre-implant CT scans of donor lungs. Previous similar reports have been of low numbers, so this is a novel study.

The references to EVLP are irrelevant, and could be removed – this is a completely separate examination, and does not need to be compared with EVLP

It is made clear only in the discussion that the transplant was already underway when the CT was performed, so the results did not alter the decision to proceed. This information should come earlier in the manuscript

We are given results only in the cases undergoing CT scan. The data should be put in the context of the overall transplant programme. How did the donor choices and recipient outcomes differ, if at all, in the lungs not subjected to CT scan? How was the decision to perform the scan made? Were there different donor features?

The donor demographics do not contain donor blood gases, a widely used marker of donor quality. Is this data available? Did it differ in lungs with infiltrates?

The data comes from over 10 years of activity. How was it distributed in this time span? Ie were a lot of the scans done in the later part of the series? This again highlights the need to put the scanned patient in the whole context of the series.

There is a simple division of those lungs with infiltrates and those without. Whilst numbers were small, was any other analysis made of the infiltrates? Were they quantified? Was there any pattern linked to any particularly less good outcome?

The microbiology results are intriguing and potentially important. How were the bronchial aspirates obtained? Do they amount to a BAL, which properly samples organisms in the lung parenchyma? If so, was this guided by the CT? Or was it just an upper airway aspiration, where the findings are, in other series, less important. We presume this was not just a suction down the endotracheal tube in the donor, so effectively a tracheal rather than bronchial aspirates?

Reviewer #2: This is a retrospective clinical study by investigators from Tohoku University in Sendai, Japan, investigating the value of ex-situ CT imaging of donor lungs immediately prior to lung transplantation.

The authors have investigated the correlation between abnormal findings on chest CT and early outcome after transplantation. They concluded that ex-situ CT imaging may help in evaluating donor lungs prior to acceptance for transplantation.

Major comments:

1) Study period:

The authors studied 70 donor lung pairs over a period of 11 years (2007-2017). It would be interesting to know how many lung transplants were performed at their centre during the study period.

- Did all donor lungs have CT imaging prior to transplantation? (70 transplants in eleven years is a very low annual number)

- If not, how were the donor lungs selected to undergo CT imaging?

- What happened to donor lungs that were already rejected for transplantation at the donor hospital? Where these lungs also recovered for CT imaging?

- please discuss the selection of donor lungs in the paper and provide a study flow chart describing all donor offers over the study period with the number of lungs recovered, lungs scanned ex-situ with CT, and lungs transplanted.

2) Terminology:

The authors have used the term “extracted donor lung (EDL) CT”. The word “extracted” is not well choosen. The reviewer would prefer the term “ex-situ” donor lung CT imaging in contrast to “in-situ” donor lung imaging while the lungs are still inside the deceased donor body.

- please change the terminology throughout the paper.

3) CT scoring and reading by radiologist:

It is well known that reading of CT images is subjective with an important inter-observer variability.

- How were the CT readers trained to describe findings of donor lungs on images taken outside the human body?

- How many radiologist were involved in the study? How many looked at the images of 1 pair of donor lungs? Was there consensus on the findings amongst radiologists?

- How did they do the scoring? Have they used a scoring system? How were the donor lungs assigned to one of both study groups?

- Were the radiologist blinded to the macroscopic findings by the surgeon who retrieved the lungs?

- Were there any lungs that were declined after reading the CT images?

- Beside “infiltration”, some of the donor lungs will have some underlying parenchymal disease (emphysema, interstitial fibrosis); Nothing is mentioned about parenchymal lung disease in the paper. See also paper by Gauthier et al (reference below). Was any underlying lung disease observed during imaging? Were these lungs excluded for transplantation?

Please describe in the methods in more details the technique and the number of radiologists involved in this study. Please discuss findings of underlying lung disease.

4) PGD grading:

The ISHLT PGD grading system was first published by Christie et al in 2005 and then modified by Snell et al in 2017. The paper by Barr M et al (reference 4) does not discuss the PGD grading system!

- Please describe which PGD grading system (2005 vs 2017) was used to score the lungs after transplantation and refer to the correct paper.

5) Sensitivity and specificity of donor X-ray and CT scan:

The authors discuss in the paper that some of the donors had in-situ imaging in the days before retrieval. It is well known that standard X-ray imaging suffers from a low sensitivity and low specificity for abnormal findings. E.g. for the BLT group, the false positive rate was 13% and the false negative was 54.5% (see table 1). The accuracy of CT scan should be better. However, the false positive rate of CT scan in this series was still 66.7% (4/6; see Table 1).

- the authors should discuss in this paper the sensitivity and specificity of standard X-ray and CT imaging based on other papers in the literature and correlate this with their own findings in the study.

- it is important to stress that the cause of a “radiologic infiltrate” in donor lungs ranges from “pneumonia” to simple “atelectasis”. Abnormal findings on the CT scan should always be checked inside the donor prior to declining a donor offer!! (see paper by Martens et al Eur J Cardiothorac Surg. 2016 Nov;50(5):832-833; Accepting donor lungs for transplant: let Lisa and Bob finish the job!)

6) References:

The authors briefly discuss 4 papers (out of a total of only 10 references) whereby ex-situ MRI was used to correlate findings. These papers do not discuss the use of CT and do no refer to lung transplantation. The reviewer believes that all 4 references should be deleted.

On the other hand, there are more papers discussing the use of CT scan to evaluate donor lungs in-situ and ex-situ!

- please discuss the following papers in the revised version:

Real-Time Computed Tomography Highlights Pulmonary Parenchymal Evolution During Ex Vivo Lung Reconditioning. Sage E, De Wolf J, Puyo P, Bonnette P, Glorion M, Salley N, Roux A, Liu N, Chapelier A. Ann Thorac Surg. 2017 Jun;103(6):e535-e537. doi: 10.1016/j.athoracsur.2016.12.02

Chest computed tomography imaging improves potential lung donor assessment. Gauthier JM, Bierhals AJ, Liu J, Balsara KR, Frederiksen C, Gremminger E, Hachem RR, Witt CA, Trulock EP, Byers DE, Yusen RD, Aguilar PR, Marklin G, Nava RG, Kozower BD, Pasque MK, Meyers BF, Patterson GA, Kreisel D, Puri V. J Thorac Cardiovasc Surg. 2019 Apr;157(4):1711-1718.e1. doi: 10.1016/j.jtcvs.2018.11.038.

Commentary: A plea for a donor CT! Hoetzenecker K. J Thorac Cardiovasc Surg. 2019 Apr;157(4):1720-1721. doi: 10.1016/j.jtcvs.2018.12.039. Epub 2018 Dec 21.

Minor comments:

7) Figure legends:

- Figure 2 and Figure 3: please correct the word “transactional” into “transaxial”

- Figure 6 B: ICU stay was not significantly different in SLT! Please correct the word “significantly” in the sentence!

- Figure 7: please refer in the legend to the correct reference for PGD grading.

8) Tables 7& 8:

Both tables can be deleted; the data can be summarized in the text in the results section.

9) Figure 7:

The current figure is difficult to interpret the differences between both groups. The reviewer believes that a survival curve showing “freedom from PGD3” would better depict the differences over time (T0-T72) between the control and the infiltration group.

Reviewer #3: The objective of this study was to investigate the correlation between EDL-CT findings and post-transplant outcome in patients who underwent bilateral lung transplantation (BLT) or single lung transplantation (SLT).

Below are some comments:

1. The nature of the study was exploratory. Allocation of control or infiltration group was not random and might depend on some variables which might or might not be directly recorded. Was there any methods applied to take into account such heterogeneity? Methods to compare the 2 groups with adjustments should be considered.

2. Lines 234-235: As acknowledge by the authors due to the non-intervention nature, this study could not demonstrate the real practical usefulness of EDL-CT. How would you use the current information on EDL-CT from this study? Is there any plan for future research?

3. In the article, it’s not clear what was the primary endpoint for the study. Given the small sample sizes in both cohorts, the test statistics and the observed p-values should be interpreted carefully (including statements on significant difference etc). Apart from PGD and ICU mortality, has other endpoints such as a 30 mortality rate had ever been considered?

4. How was the sample size and the power for this study determined? This should be clarified. If it is a convenient sample size, reasons to support such case should be underlined.

5. The PGD grade at each time point was analyzed by repeated measures two-way ANOVA. As in point 1, please explain how the heterogeneity can be handled with this technique. Do you consider an alternative model to take into account heterogeneity of the different baseline factors when conducting this analysis?

6. Lines: 226-227: ‘CT. We believe that EDL-CT may predict the post-transplant outcome in lung transplantation.’ Please provide evidence for this statement. Do you have any supporting data in the current study?

6. PLOS authors have the option to publish the peer review history of their article (what does this mean?). If published, this will include your full peer review and any attached files.

Reviewer #1: Yes: John H Dark

Reviewer #2: Yes: Dirk Van Raemdonck

Reviewer #3: No

---

## [Author Response · Author response to Decision Letter 0]

24 Apr 2020

Details of our revisions are outlined in a point-by-point response to the reviewers. Please see the file "Response to Reviewers."

---

## [Decision Letter · Decision Letter 1]

8 May 2020

PONE-D-20-03435R1

Ex vivo lung CT findings may predict the outcome of the early phase after lung transplantation

PLOS ONE

Dear Dr. Oishi,

Thank you for submitting your manuscript to PLOS ONE. After careful consideration, we feel that it has merit but does not fully meet PLOS ONE’s publication criteria as it currently stands. Therefore, we invite you to submit a revised version of the manuscript that addresses the points raised during the review process.

Thank you for making the changes as requested. I do agree with reviewer 3 to include a statement about the lack of sample size calculation, and to add a CONSORT diagram (patient seleciton/study groups). This will make the MS much more clear.

We would appreciate receiving your revised manuscript by Jun 22 2020 11:59PM. To enhance the reproducibility of your results, we recommend that if applicable you deposit your laboratory protocols in protocols.io, where a protocol can be assigned its own identifier (DOI) such that it can be cited independently in the future. For instructions see: http://journals.plos.org/plosone/s/submission-guidelines#loc-laboratory-protocols

We look forward to receiving your revised manuscript.

Kind regards,

Frank JMF Dor, M.D., Ph.D., FEBS, FRCS

Academic Editor

PLOS ONE

Reviewers' comments:

Reviewer's Responses to Questions

**Comments to the Author**

1. If the authors have adequately addressed your comments raised in a previous round of review and you feel that this manuscript is now acceptable for publication, you may indicate that here to bypass the “Comments to the Author” section, enter your conflict of interest statement in the “Confidential to Editor” section, and submit your "Accept" recommendation.

Reviewer #2: All comments have been addressed

Reviewer #3: (No Response)

2. Is the manuscript technically sound, and do the data support the conclusions?

Reviewer #2: Yes

Reviewer #3: Yes

3. Has the statistical analysis been performed appropriately and rigorously? 

Reviewer #2: Yes

Reviewer #3: Yes

4. Have the authors made all data underlying the findings in their manuscript fully available?

Reviewer #2: Yes

Reviewer #3: Yes

5. Is the manuscript presented in an intelligible fashion and written in standard English?

Reviewer #2: Yes

Reviewer #3: Yes

6. Review Comments to the Author

Reviewer #2: The reviewer would like to thank the authors for replying to all my comments and for changing the revised manuscript accordingly.

No further comments or questions.

Reviewer #3: Thanks for addressing the comments. It is understood that the study is exploratory in nature and there is no sample size calculation performed. I would suggest that you include that in the article. It is important to add a CONSORT diagram of patient selection and study groups. This will make the article more readable and explain the complexity of the patient selection into study.

7. PLOS authors have the option to publish the peer review history of their article (what does this mean?). If published, this will include your full peer review and any attached files.

Reviewer #2: No

Reviewer #3: No

---

## [Author Response · Author response to Decision Letter 1]

12 May 2020

Please see the file "Response to Reviewers-ver.3."

---

## [Editor Report · Decision Letter 2]

13 May 2020

Ex vivo lung CT findings may predict the outcome of the early phase after lung transplantation

PONE-D-20-03435R2

Dear Dr. Oishi,

We are pleased to inform you that your manuscript has been judged scientifically suitable for publication and will be formally accepted for publication once it complies with all outstanding technical requirements.

With kind regards,

Frank JMF Dor, M.D., Ph.D., FEBS, FRCS

Academic Editor

PLOS ONE

Additional Editor Comments (optional):

thank you for making the suggested changes to the MS.
---

## [Editor Report · Acceptance letter]

20 May 2020

PONE-D-20-03435R2 

Ex vivo lung CT findings may predict the outcome of the early phase after lung transplantation 

Dear Dr. Oishi:

I am pleased to inform you that your manuscript has been deemed suitable for publication in PLOS ONE. Congratulations! Your manuscript is now with our production department. 

With kind regards,

on behalf of

Dr. Frank JMF Dor 

Academic Editor

PLOS ONE